# Accumulation Bit-Width Scaling For Ultra-Low Precision Training Of Deep Networks

**Charbel Sakr**[†‡*]**, Naigang Wang**[‡]**, Chia-Yu Chen**[‡]**, Jungwook Choi**[‡]**, Ankur Agrawal**[‡]**,
Naresh Shanbhag**[†]**, Kailash Gopalakrishnan**[‡]
[†] Dept. of Electrical and Computer Engineering, University of Illinois at Urbana-Champaign
`{sakr2,shanbhag}@illinois.edu`
[‡] IBM T.J. Watson Research Center
`{nwang,cchen,choij,ankuragr,kailash}@us.ibm.com`

## Abstract

Efforts to reduce the numerical precision of computations in deep learning training have yielded systems that aggressively quantize weights and activations, yet employ wide high-precision accumulators for partial sums in inner-product operations to preserve the quality of convergence. The absence of any framework to analyze the precision requirements of partial sum accumulations results in conservative design choices. This imposes an upper-bound on the reduction of complexity of multiply-accumulate units. We present a statistical approach to analyze the impact of reduced accumulation precision on deep learning training. Observing that a bad choice for accumulation precision results in loss of information that manifests itself as a reduction in variance in an ensemble of partial sums, we derive a set of equations that relate this variance to the length of accumulation and the minimum number of bits needed for accumulation. We apply our analysis to three benchmark networks: CIFAR-10 ResNet 32, ImageNet ResNet 18 and ImageNet AlexNet. In each case, with accumulation precision set in accordance with our proposed equations, the networks successfully converge to the single precision floating-point baseline. We also show that reducing accumulation precision further degrades the quality of the trained network, proving that our equations produce tight bounds. Overall this analysis enables precise tailoring of computation hardware to the application, yielding area- and power-optimal systems.

## 1 Introduction

Over the past decade, deep learning techniques have been remarkably successful in a wide spectrum of applications through the use of very large and deep models trained using massive datasets. This training process necessitates up to 100's of ExaOps of computation and Gigabytes of storage. It is, however, well appreciated that a range of approximate computing techniques can be brought to bear to significantly reduce this computational complexity (Chen et al., 2018) - and amongst them, exploiting reduced numerical precision during the training process is extremely effective and has already been widely deployed (Gupta et al., 2015).

There are several reasons why reduced precision deep learning has attracted the attention of both hardware and algorithms researchers. First, it offers well defined and scalable hardware efficiency, as opposed to other complexity reduction techniques such as pruning (Han et al., 2015b;a), where handling sparse data is needed. Indeed, parameter complexity scales linearly while multiplication hardware complexity scales quadratically with precision bit-width (Zhou et al., 2016). Thus, any advance towards truly binarized networks (Hubara et al., 2016) corresponds to potentially 30x - 1000x complexity reduction in comparison to single precision floating-point hardware. Second, the mathematics of reduced precision has direct ties with the statistical theory of quantization (Widrow and Kollár, 2008). In the context of deep learning, this presents an opportunity for theoreticians to derive analytical trade-offs between model accuracy and numerical precision (Lin et al., 2016; Sakr et al., 2017).

---

[*]Work done while at IBM Research.

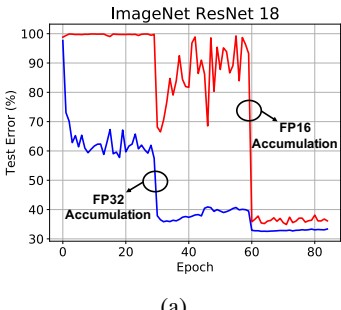 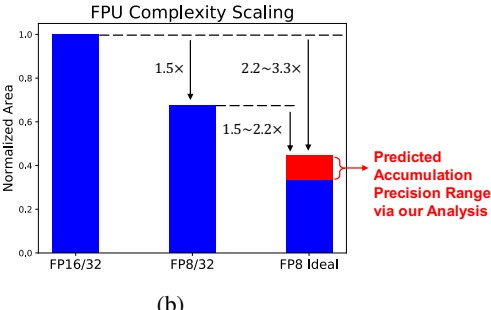

(a)                  (b)

Figure 1: The importance of accumulation precision: (a) convergence curves of an ImageNet ResNet 18 experiment using reduced precision accumulation. The current practice is to keep the accumulation in full precision to avoid such divergence. (b) estimated area benefits when reducing the precision of a floating-point unit (FPU). The terminology FP$a/b$ denotes an FPU whose multiplier and adder use $a$ and $b$ bits, respectively. Our work enables convergence in reduced precision accumulation and gains an extra $1.5\times \sim 2.2\times$ area reduction.

Most ongoing efforts on reduced precision deep learning solely focus on quantizing representations and always assume wide accumulators, i.e., ideal summations. The reason being reduced precision accumulation can result in severe training instability and accuracy degradation, as illustrated in Figure 1 (a) for ResNet 18 (ImageNet) model training. This is especially unfortunate, since the hardware complexity in reduced precision floating-point numbers (needed to represent small gradients during training) (Wang et al., 2018; Micikevicius et al., 2017) is dominated by the accumulator bit-width. To illustrate this dominance we developed a model underpinned by the hardware synthesis of low-precision floating-point units (FPU), that translates precision into area complexity of the FPU. Comparisons obtained from this model are shown in Figure 1 (b). We observe that accumulating in high precision severely limits the hardware benefits of reduced precision computations. This presents a new angle to the problem of reduced precision deep learning training which concerns determining suitable accumulation precision and forms the basis of our paper. Our findings are that the accumulation precision requirements in deep learning training are nowhere near 32-b, and in fact could enable further complexity reduction of FPUs by a factor of $1.5 \sim 2.2\times$.

## 1.1 RELATED WORKS

Our work is concerned with establishing theoretical foundations for estimating the accumulation bit precision requirements in deep learning. While this topic has never been addressed in the past, there are prior works in both deep learning and high performance computing communities that align well with ours.

Most early works on reduced precision deep learning consider fixed-point arithmetic or a variation of it (Gupta et al., 2015). However, when considering quantization of signals involving the back-propagation algorithm, finding a suitable fixed-point configuration becomes challenging due to a weak handle on the scalar dynamic range of the back-propagated signals. Thus, hardware solutions have been sought, and, accordingly, other number formats were considered. Flexpoint (Köster et al., 2017) is a hybrid version between fixed-point and floating-point where scalars in a tensor are quantized to 16-b fixed-point but share 5-b of exponent to adjust the dynamic range. Similarly, WAGE (Wu et al., 2018) augments Flexpoint with stochastic quantization and enables integer quantization. All of these schemes focused on representation precision, but mostly used 32-bit accumulation. Another option is to use reduced precision floating-point as was done in MPT (Micikevicius et al., 2017), which reduces the precision of most signals to 16-b floating-point, but observes accuracy degradation when reducing the accumulation precision from 32-b. Recently, Wang et al. (2018) quantize all representations to 8-b floating-point and experimentally find that the accumulation could be in 16-b with algorithmic contrivance, such as chunk-based accumulation, to enable convergence.

The issue of numerical errors in floating-point accumulation has been classically studied in the area of high performance computing. Robertazzi and Schwartz (1988) were among the first to statistically estimate the effects of floating-point accumulation. Assuming a stream of uniformly and exponentially distributed positive numbers, estimates for the mean square error of the floating-

point accumulation were derived via quantization noise analysis. Because such analyses are often intractable (due to the multiplicative nature of the noise), later works on numerical stability focus on worst case estimates of the accumulation error. Higham (1993) provide upper bounds on the error magnitude by counting and analyzing round-off errors. Following this style of worst case analysis, Castaldo et al. (2008) provide bounds on the accumulation error for different summing algorithms, notably using chunk-based summations. Different approaches to chunking are considered and their benefits are estimated. It is to be noted that these analyses are often loose as they are agnostic to the application space. To the best of our knowledge, a statistical analysis on the accumulation precision specifically tailored to deep learning training remains elusive.

### 1.2 CONTRIBUTIONS

Our contribution is both theoretical and practical. We introduce the variance retention ratio (VRR) of a reduced precision accumulation in the context of the three deep learning dot products. The VRR is used to assess the suitability, or lack thereof, of a precision configuration. Our main result is the derivation of an actual formula for the VRR that allows us to determine accumulation bit-width for precise tailoring of computation hardware. Experimentally, we verify the validity and tightness of our analysis across three benchmarking networks (CIFAR-10 ResNet 32, ImageNet ResNet 18 and ImageNet AlexNet).

## 2 BACKGROUND ON FLOATING-POINT ARITHMETIC

The following basic floating-point definitions and notations are used in our work:

**Floating-point representation:** A $b$-bit floating-point number $a$ has a signed bit, $e$ exponent bits, and $m$ mantissa bits so that $b = 1 + e + m$. Its binary representation is $(B_s, B'_1, \ldots, B'_e, B''_1, \ldots, B''_m) \in \{0,1\}^b$ and its value is equal to: $a = (-1)^{B_s} \times 2^E \times (1 + M)$ where $E = -(2^{e-1} - 1) + \sum_{i=1}^{e} B'_i 2^{(e-i)}$ and $M = \sum_{i=1}^{m} B''_i 2^{-i}$. Such number is called a $(1, e, m)$ floating-point number.

**Floating-point operations:** One of the most pervasive arithmetic functions used in deep learning is the dot product between two vectors which is the building block of the generalized matrix multiplication (GEMM). A dot product is computed in a multiply-accumulate (MAC) fashion and thus requires two floating-point operations: multiplication and addition. The realization of an ideal floating-point operation requires a certain bit growth at the output to avoid loss of information. For instance, in a typical MAC operation, if $c \leftarrow c + a \times b$ where $a$ is $(1, e_a, m_a)$ and $b$ is $(1, e_b, m_b)$, then $c$ should be $(1, \max(e_a, e_b) + 2, m_a + m_b + 1 + \Delta_E)$, which depends on the bit-precision and the relative exponent difference of the operands $\Delta_E$. However, it is often more practical to pre-define the precision of c as $(1, e_c, m_c)$, which requires rounding immediately after computation. Such rounding might cause an operand to be completely or partially truncated out of the addition, a phenomenon called "swamping" (Higham, 1993), which is the primary source of accumulation errors.

## 3 ACCUMULATION VARIANCE

The second order statistics (variance) of signals are known to be of great importance in deep learning. For instance, in prior works on weight initialization (Glorot and Bengio, 2010; He et al., 2015), it is customary to initialize random weights subject to a variance constraint designed so as to prevent vanishing or explosion of activations and gradients. Thus, such variance engineering induces fine convergence of DNNs. Importantly, in such analyses, the second order output statistics of a dot product are studied and expressed as a function of that of the accumulated terms, which are assumed to be independent and having similar variance. A fundamental assumption is: $Var(s) = nVar(p)$, where $Var(s)$ and $Var(p)$ are the variances of the sum and individual product terms, respectively, and $n$ is the length of the dot product. One intuition concerning accumulation with reduced precision is that, due to swamping, some product terms may vanish from the summation, resulting in a lower variance than expected: $Var(s) = \tilde{n}Var(p)$, where $\tilde{n} < n$. This constitutes a violation of a key assumption and effectively leads to the re-emergence of the difficulties in training neural networks with improper weight initialization which often harms the convergence behavior (Glorot and Bengio, 2010; He et al., 2015).

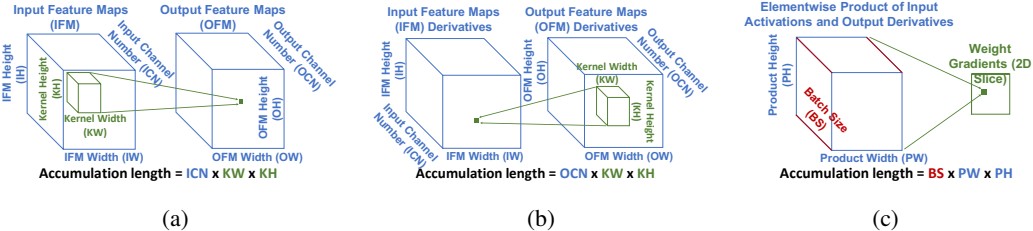

(a)  (b)  (c)

Figure 2: The three GEMM calls, and hence accumulations, in one iteration of the back-propagation algorithm: (a) the forward propagation (FWD), (b) the backward propagation (BWD), and (c) gradient computation (GRAD). The accumulation of these three GEMMs is across multiple dimensions (mini-batch size, feature maps, output channels etc.) and their lengths are usually very long.

To explain the poor convergence of our ResNet 18 run (Figure 1 (a)), we evaluate the behavior of accumulation variance across layers. Specifically, we check the three dot products of a back-propagation iteration: the forward propagation (FWD), the backward propagation (BWD), and the gradient computation (GRAD), as illustrated in Figure 2. Indeed, there is an abnormality in reduced precision GRAD as shown in Figure 3. It is also observed that the abnormality of variance is directly related to accumulation length. From Figure 3, the break point corresponds to the switch from the first to the second residual block. The GRAD accumulation length in the former is much longer ($4\times$) than the latter. Thus, evidence points to the direction that for a given precision, there is an accumulation length for which the expected variance cannot be properly retained due to swamping. Motivated by these observations, we propose to study the trade-offs among accumulation variance, length, and mantissa precision.

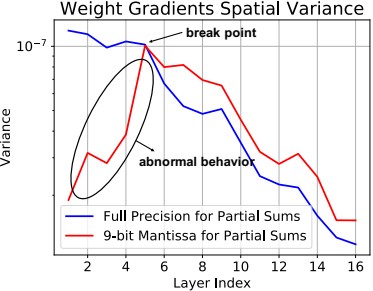

Figure 3: Snapshot of measured weight gradient variance as a function of layer index for our ResNet 18 experiment. An abnormal variance is observed for the reduced precision accumulation.

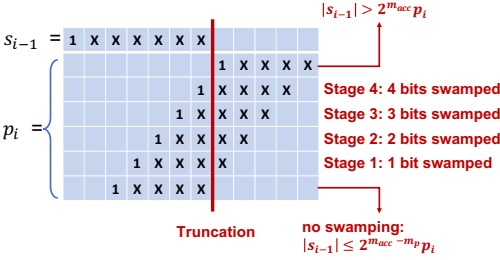

Figure 4: Illustration of the difference between full swamping and partial swamping when $m_{acc} = 6$ and $m_p = 4$. The bit-shift of $p_i$ due to exponent difference may cause partial (e.g., stage 1-4) or full truncation of $p_i$'s bits, called swamping.

Before proceeding, it is important to note that our upcoming analysis differs, in style, from many works on reduced precision deep learning where it is common to model quantization effects as additive noise causing increased variance (McKinstry et al., 2018). Our work does not contradict such findings, since prior arts have considered representation quantization whose effects are, by nature, different from intermediate roundings in partial sums.

## 4 MANTISSA PRECISION REQUIREMENTS ANALYSIS

We assume sufficient exponent precision throughout and treat reduced precision floating-point arithmetic as an unbiased form of approximate computing, as is customary. Thus, our work focuses on associating second order statistics to mantissa precision.

We consider the accumulation of $n$ terms $\{p_i\}_{i=1}^{n}$ which correspond to the element-wise product terms in a dot product. The goal is to compute the correct $n^{\text{th}}$ partial sum $s_n$ where $s_i = \sum_{i'=1}^{i} p_{i'}$.

We assume, as in (He et al., 2015), that the product terms $\{p_i\}_{i=1}^n$ are statistically independent, zero-mean, and have the same variance $\sigma_p^2$. Thus, under ideal computation, the variance of $s_n$ (which is equal to its second moment) should be $Var(s_n)_{\text{ideal}} = n\sigma_p^2$. However, due to swamping effects, the variance of $s_n$ under reduced precision is $Var(s_n)_{\text{swamping}} \neq Var(s_n)_{\text{ideal}}$.

Let the product terms $\{p_i\}_{i=1}^n$ and partial sum terms $\{s_i\}_{i=1}^n$ have $m_p$ and $m_{acc}$ mantissa bits, respectively. Our key contribution is a formula for the *variance retention ratio* $VRR = \frac{Var(s_n)_{\text{swamping}}}{Var(s_n)_{\text{ideal}}}$. The VRR, which is always less than or equal to unity, is a function of $n$, $m_p$, and $m_{acc}$ only, which needs no simulations to be computed. Furthermore, to preserve quality of computation under reduced precision, it is required that $VRR \to 1$. As it turns out, the VRR for a fixed precision is a curve with "knee" with respect to $n$, where a break point in accumulation length beyond which a certain mantissa precision is no longer suitable can be easily identified. Accordingly, for a given accumulation length, the mantissa precision requirements can be readily estimated.

Before proceeding, we formally define swamping. As illustrated in Figure 4, the bit-shift of $p_i$ due to exponent difference may cause partial (e.g., stage 1-4) or full truncation of $p_i$'s bits, called swamping. We define (1) "full swamping" which occurs when $|s_i| > 2^{m_{acc}}|p_{i+1}|$, and (2) "partial swamping" which occurs when $2^{m_{acc}-m_p}|p_{i+1}| < |s_i| \leq 2^{m_{acc}}|p_{i+1}|$. These two swamping types will be fully considered in our analysis.

### 4.1 Variance Retention Ratio

In the lemma below, we first present a formula for the VRR when only full swamping is considered.

**Lemma 1.** *The variance retention ratio of a length $n$ accumulation using $m_{acc}$ mantissa bits, and when only considering full swamping, is given by:*

$$VRR_{\text{full swamping}} = \frac{\sum_{i=2}^{n-1} iq_i + n\tilde{q}_n}{kn} \tag{1}$$

*where* $q_i = 2Q\left(\frac{2^{m_{acc}}}{\sqrt{i}}\right)\left(1 - 2Q\left(\frac{2^{m_{acc}}}{\sqrt{i-1}}\right)\right)$, $\tilde{q}_n = 1 - 2Q\left(\frac{2^{m_{acc}}}{\sqrt{n}}\right)$, $k = \sum_{i=2}^{n-1} q_i + \tilde{q}_n$ *is a normalization constant, and $Q$ denotes the elementary Q-function.*

The proof is provided in Appendix A. A preliminary check is that a very large value of $m_{acc}$ in (1) causes all $\{q_i\}_{i=1}^{n-1}$ terms to vanish and $\tilde{q}_n$ to approach unity. This makes $VRR \to 1$ for high precision as expected. On the other hand, if we assume $m_{acc}$ to be small, but let $n \to \infty$, we get $n\tilde{q}_n \to 0$ because the Q-function term will approach 1 exponentially fast as opposed to the $n$ term which is linear. Furthermore, the terms inside the summation having a large $i$ will vanish by the same argument, while the $n$ term in the denominator will make the ratio decrease and we would expect $VRR \to 0$. This means that with limited precision, there is little hope to achieve a correct result when the accumulation length is very large. Also, the rapid change in VRR from 0 to 1 indicates that VRR can be used to provide sharp decision boundary for accumulation precision.

The above result only considers full swamping and is thus incomplete. Next we augment our analysis to take into account the effects of partial swamping. The corresponding formula for the VRR is provided in the following theorem.

**Theorem 1.** *The variance retention ratio of a length $n$ accumulation using $m_p$ and $m_{acc}$ mantissa bits for the input products and partial sum terms, respectively, is given by:*

$$VRR = \frac{\sum_{i=2}^{n-1}(i-\alpha)_+ q_i \mathbf{1}_{\{i>\alpha\}} + \sum_{j_r=2}^{m_p}(n-\alpha_{j_r})_+ q'_i \mathbf{1}_{\{n>\alpha_{j_r}\}} + nk_3}{kn} \tag{2}$$

*where* $(x)_+ = \begin{cases} x \text{ if } x > 0 \\ 0 \text{ otherwise} \end{cases}$, $\mathbf{1}_A = \begin{cases} 1 \text{ if } A \text{ is true} \\ 0 \text{ otherwise} \end{cases}$,

$\alpha = \frac{2^{m_{acc}-3m_p}}{3}\sum_{j=1}^{m_p} 2^j(2^j-1)(2^{j+1}-1)$, $q_i = 2Q\left(\frac{2^{m_{acc}}}{\sqrt{i}}\right)\left(1 - 2Q\left(\frac{2^{m_{acc}}}{\sqrt{i-1}}\right)\right)$,

$\alpha_{j_r} = \frac{2^{m_{acc}-3m_p}}{3}\sum_{j=1}^{j_r-1} 2^j(2^j-1)(2^{j+1}-1)$,

$q'_{j_r} = N_{j_r-1}2Q\left(\frac{2^{m_{acc}-m_p+j_r-1}}{\sqrt{n}}\right)\left(1 - 2Q\left(\frac{2^{m_{acc}-m_p+j_r}}{\sqrt{n}}\right)\right)$, $k = k_1 + k_2 + k_3$,

$k_1 = \sum_{i=2}^{n-1} q_i \mathbf{1}_{\{i>\alpha\}}$, $k_2 = \sum_{j_r=2}^{m_p} q'_i \mathbf{1}_{\{n>\alpha_{j_r}\}}$, *and* $k_3 = 1 - 2Q\left(\frac{2^{m_{acc}-m_p+1}}{\sqrt{n}}\right)$.

The proof is provided in Appendix B. Observe the dependence on $m_{acc}$, $m_p$, and $n$. Therefore, in what follows, we shall refer to the VRR in (2) as $VRR(m_{acc}, m_p, n)$. Once again, we verify the extremal behavior of our formula. A very large value of $m_{acc}$ in (2) causes $k_1 \approx k_2 \approx 0$ and $k_3 \approx 1$. This makes $VRR \to 1$ for high precision as expected. In addition, assuming small $m_{acc}$ and letting $n \to \infty$, we get $nk_3 \to 0$ because $k_3$ decays exponentially fast due to the Q-function term. By the same argument, $q'_{j_r} \to 0$ for all $j_r$ and $q_i \to 0$ for all but small values of $i$. Thus, the numerator will be small, while the denominator will increase linearly in $n$ causing $VRR \to 0$. Thus, once more, we establish that with limited accumulation precision, there is little hope for a correct result.

## 4.2 VRR with Chunk Based Accumulations

Next we consider an accumulation that uses chunking. In particular, assume $n = n_1 \times n_2$ so that the accumulation is broken into $n_2$ chunks, each of length $n_1$. Thus, $n_2$ accumulations of length $n_1$ are performed and the $n_2$ intermediate results are added to obtain $s_n$. This simple technique is known to greatly improve the stability of sums (Castaldo et al., 2008). The VRR can be used to theoretically explain such improvements. For simplicity, we assume two-level chunking (as described above) and same mantissa precision $m_{acc}$ for both inter-chunk and intra-chunk accumulations. Applying the above analysis, we may obtain a formula for the VRR as provided in the corollary below, which is proved in Appendix C.

**Corollary 1.** *The variance retention ratio of an length $n = n_1 \times n_2$ accumulation with chunking, where $n_1$ is the chunk size and $n_2$ is the number of chunks, using $m_p$ and $m_{acc}$ mantissa bits for the input products and partial sum terms, respectively, is given by:*

$$VRR_{chunking} = VRR(m_{acc}, m_p, n_1) \times VRR\left(m_{acc}, \min\left(m_{acc}, m_p + \log_2(n_1)\right), n_2\right) \quad (3)$$

## 4.3 VRR with Sparsity

It is common to encounter sparse operands in deep learning dot products. Since addition of zero is an identity operation, the effective accumulation length is often less than as described by the network topology. Indeed, for a given accumulation, supposedly of length $n$, if we can estimate the non-zero ratio (NZR) of its incoming product terms, then the effective accumulation length is $NZR \times n$.

Thus, when an accumulation is known to have sparse inputs with known NZR, a better estimate of the VRR is

$$VRR_{\text{sparsity}} = VRR(m_{acc}, m_p, NZR \times n). \quad (4)$$

Similarly, when considering the VRR with chunking, we may use knowledge of sparsity to obtain the effective intra-accumulation length as $NZR \times n_1$. This change is reflected both in the VRR of the intra-chunk accumulation and the input precision of the inter-chunk accumulation:

$$
\begin{aligned}
VRR_{\text{chunking and sparsity}} = {} & VRR(m_{acc}, m_p, NZR \times n_1) \\
& \times VRR\left(m_{acc}, \min\left(m_{acc}, m_p + \log_2(NZR \times n_1)\right), n_2\right) \quad (5)
\end{aligned}
$$

In practice, the NZR can be estimated by making several observations from baseline data. Using an estimate of the NZR makes our analysis less conservative.

## 4.4 Usage of Analysis

For a given accumulation setup, one may compute the VRR and observe how close it is from the ideal value of 1 in order to judge the suitability of the mantissa precision assignment. It turns out that when measured as a function of accumulation length $n$ for a fixed precision, the VRR has a breakdown region. This breakdown region can very well be observed when considering what we define as the normalized exponential variance lost:

$$v(n) = e^{n(1-VRR)} \quad (6)$$

In Figure 5 (a,b) we plot $v(n)$ for different values of $m_{acc}$ when considering both normal accumulation and chunk-based accumulation with a chunk-size of 64. The value of $m_p$ is set to 5-b, corresponding to the product of two numbers in (1,5,2) floating-point format (Wang et al., 2018). We consider $m_{acc}$ to be suitable for a given $n$ only if $v(n) < 50$. The reason being, in all plots, the

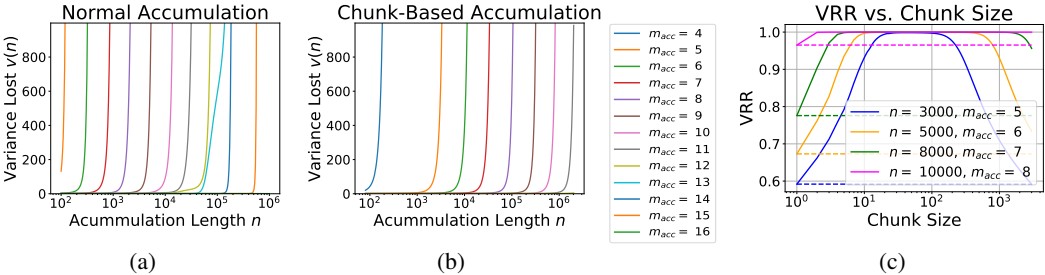

Figure 5: Normalized variance lost as a function of accumulation length for different values of $m_{acc}$ for (a) a normal accumulation (no chunking) and (b) a chunk-based accumulation (chunk size of 64). The "knees" in each plot correspond to the maximum accumulation length for a given precision which indicates how the VRR is to be used to select a suitable precision. (c) VRR as a function of chunk-size for several accumulation setups. The dashed lines correspond to the value of the VRR when no chunking is used. The flat maximas indicate that the exact choice of a chunking size is not of paramount importance.

variance lost rapidly increases when $v(n) > 50$ and $n$ increases. On the other hand, when $v(n) < 50$ and $n$ decreases, the variance lost quickly drops to zero. This choice of a cut-off value is thus chosen purely based on the accumulation length and precision.

In addition, when performing chunk-based accumulation, the chunk size is a hyperparameter that, a priori, cannot be determined trivially. Castaldo et al. (2008) identified an optimal chunk size minimizing the loose upper bound on the accumulation error they derived. In practice, they did not find the accumulation error to be sensitive to the chunk-size. Neither did Wang et al. (2018) who performed numerical simulations. By sweeping the chunk size and observing the accumulation behavior on synthetic data, it was found that chunking significantly reduces accumulation error as long as the chunk size is not too small nor too large. Using our analysis, we provide a theoretical justification. Figure 5 (c) shows the VRR for various accumulation setups, including chunking when the chunk size is swept. For each case we see that chunking raises the VRR to a value close to unity. Furthermore, the VRR curve in that regime is "flat", meaning that a specific value of chunk size does not matter as long as it is not too small nor too large. One intuition is that a moderate chunk size prevents both inter- and intra-chunk accumulations to be as large as the original accumulation. In our upcoming chunking experiments we use a chunk size of 64 as was done by Wang et al. (2018).

## 5 NUMERICAL RESULTS

Using the above analysis, we predict the mantissa precisions required by the three GEMM functions for training the following networks: ResNet 32 on the CIFAR-10 dataset, ResNet 18 and AlexNet on the ImageNet dataset. Those benchmarks were chosen due to both their popularity and topologies which present large accumulation lengths, making them good candidates against which we can verify our work. We use the same configurations as (Wang et al., 2018), in particular, we use 6-b of exponents in the accumulations, and quantize the intermediate tensors to (1,5,2) floating-point format and keep the final layer's precision in 16 bit. The technique of loss scaling (Micikevicius et al., 2017) is used in order to limit underflows of activation gradients. A single scaling factor of 1000 was used for all models tested.

In order to realize rounding of partial sums, we modify the CUDA code of the GEMM function (which, in principle, can be done using any framework). In particular, we add a custom rounding function where the partial sum accumulation occurs. Quantization of dot product inputs is handled similarly.

The predicted precisions for each network and layer/block are listed in Table 1 for the case of normal and chunk-based accumulation with a chunk size of 64. Several insights are to be noted.

- The required accumulation precision for CIFAR-10 ResNet 32 is in general lower than that of the ImageNet networks. This is simply because, the network topology imposes shorter dot products.
- Though, topologically, the convolutional layers in the two ImageNet networks are similar, the precision requirements do vary. Specifically, the GRAD accumulation depends on the feature map

**CIFAR-10 ResNet 32**

| Layer(s) | Conv 0 | ResBlock 1 | ResBlock 2 | Resblock 3 |
|---|---|---|---|---|
| FWD | (6,5) | (6,5) | (7,5) | (7,5) |
| BWD | N/A | (6,5) | (7,5) | (8,5) |
| GRAD | (11,8) | (11,8) | (10,6) | (9,6) |

**ImageNet ResNet 18**

| Layer(s) | Conv 0 | ResBlock 1 | ResBlock 2 | Resblock 3 | ResBlock 4 |
|---|---|---|---|---|---|
| FWD | (9,6) | (7,5) | (8,5) | (8,5) | (9,6) |
| BWD | N/A | (8,6) | (9,6) | (9,6) | (10,6) |
| GRAD | (15,10) | (15,9) | (12,8) | (10,6) | (9,5) |

**ImageNet AlexNet**

| Layer | Conv 1 | Conv 2 | Conv 3 | Conv 4 | Conv 5 | FC 1 | FC 2 |
|---|---|---|---|---|---|---|---|
| FWD | (7,5) | (9,5) | (9,5) | (8,5) | (8,5) | (9,6) | (8,5) |
| BWD | N/A | (8,5) | (8,5) | (10,8) | (8,5) | (8,5) | (8,5) |
| GRAD | (10,7) | (9,6) | (8,6) | (6,5) | (6,5) | (6,5) | (6,5) |

Table 1: The predicted precisions required for all accumulations of our considered networks. Each table entry is an ordered tuple of two values which correspond to the predicted mantissa precision of both normal and chunk-based accumulations, respectively. The precision requirements of FWD and BWD are typically smaller than those of GRAD. The latter needs the most precision for layers/blocks close to the input as the size of the feature maps is highest in the early stages. The benefits of chunking are non linear but range from 1 to 6 bits.

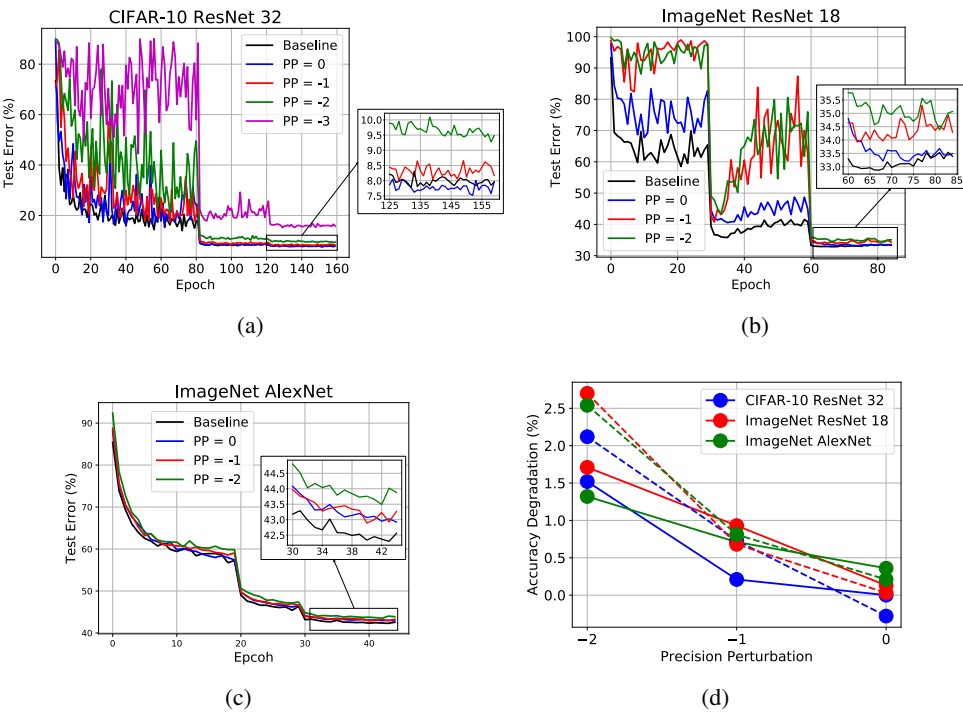

(a)

(b)

(c)

(d)

Figure 6: Convergence curves for (a) CIFAR-10 ResNet 32, (b) ImageNet ResNet 18, and (c) ImageNet AlexNet, and (d) final accuracy degradation with respect to the baseline as a function of precision perturbation (PP). The solid and dashed lines correspond to the no chunking and chunking case, respectively. Using our predicted precision assignment, the converged test error is close to the baseline (no more than 0.5% degradation) but increases significantly when the precision is further reduced.

dimension which is mostly dataset dependent, yet AlexNet requires less precision than ResNet 18. This is because the measured sparsity of the operands was found to be much higher for AlexNet.

- Figure 5 suggests that chunking decreases the precision requirements significantly. This is indeed observed in Table 1, where we see that the benefits of chunking reach up to 6-b in certain accumulations, e.g., the GRAD acccumulation in the first ResBlock of ImageNet ResNet 18.

Because our predicted precision assignment ensures the VRR of all GEMM accumulations to be close to unity, we expect reduced precision training to converge with close fidelity to the baseline. Since our work focuses on accumulation precision, in our experiments, the baseline denotes accumulation in full precision. For a fair comparison, all upcoming results use (1,5,2) representation precision. Thus, the effects of reduced precision representation are not taken into account.

The goal of our experiments is to investigate both the validity and conservatism of our analysis. In Figure 6, we plot the convergence curves when training with our predicted accumulation precision for a normal accumulation. The runs corresponding to chunk-based accumulations were also performed but are omitted since the trend is similar. Furthermore, we repeat all experiments with precision perturbation (PP), meaning a specific reduction in precision with respect to our prediction. For instance, $PP = 0$ indicates our prediction while $PP = -1$ corresponds to a one bit reduction. Finally, in order to better visualize how the accumulation precision affect convergence, we plot in Figure 6 (d) the accuracy degradation as a function of precision perturbation for each of our three networks with both normal and chunk-based accumulations. The following is to be noted:

- When $PP = 0$, the converged accuracy always lies within 0.5% of the baseline, a strong indication of the validity of our analysis. We use a 0.5% accuracy cut-off with respect to the baseline as it corresponds to an approximate error bound for neural networks obtained by changing the random numbers seed (Goyal et al., 2017; Gastaldi, 2017).
- When $PP < 0$, a noticeable accuracy degradation is observed, most notably for ImageNet ResNet 18. The converged accuracy is no longer within 0.5% of the baseline. Furthermore, a clear trend observed is that the higher the perturbation, the worse the degradation.
- ImageNet AlexNet is more robust to perturbation than the two ResNets. While $PP = -1$ causes a degradation strictly $> 0.5\%$, it is not much worse than the $PP = 0$ case. This observation aligns with that from neural net quantization that Alexnet is robust due to its over-parameterized network structure (Zhu et al., 2016). But the trend of increasing degradation remains the same.
- Figure 6 (d) suggests that the effects of PP are more pronounced for a chunk-based accumulation. Since the precision assignment itself is lower (Table 1), a specific precision perturbation corresponds to a relatively higher change. For example, decreasing one bit from a 6-b assignment is more important than decreasing one bit from a 10-b assignment. Further justification can be obtained by comparing Figures 5 (a) and 5 (b) where consecutive lines are less closely aligned for the chunk-based accumulation, indicating more sensitivity to precision perturbation.

Thus, overall, our predictions are adequate and close to the limits beyond which training becomes unstable. These are very encouraging signs that our analysis is both valid and tight.

## 6 CONCLUSION

We have presented an analytical method to predict the precision required for partial sum accumulation in the three GEMM functions in deep learning training. Our results prove that our method is able to accurately pinpoint the minimum precision needed for the convergence of benchmark networks to the full-precision baseline. Our theoretical concepts are application agnostic, and an interesting extension would be to consider recurrent architectures such as LSTMs. In particular, training via backpropagation in time could make the GRAD accumulation very large depending on the number of past time-steps used. In such a case, our analysis is of great relevance to training precision optimization. On the practical side, this analysis is a useful tool for hardware designers implementing reduced-precision FPUs, who in the past have resorted to computationally prohibitive brute-force emulations. We believe this work addresses a critical missing link on the path to truly low-precision floating-point hardware for DNN training.

### ACKNOWLEDGMENT

This work is supported in part by IBM Research; IBM SoftLayer; IBM Cognitive Computing Cluster (CCC); IBM-ILLINOIS Center for Cognitive Computing Systems Research (C3SR) - a research collaboration as part of the IBM AI Horizons Network; and C-BRIC, one of six centers in JUMP, a Semiconductor Research Corporation (SRC) program sponsored by DARPA. The authors would like to thank I-Hsin Chung, Ming-Hung Chen and Silvia Melitta Mueller for helpful discussions and support.

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

# APPENDIX

## PRELIMINARIES AND ASSUMPTIONS

The analysis of stability of sums under reduced-precision floating-point accumulation is a classically difficult problem. Indeed, statistically characterizing recursive rounding effects is often mathematically intractable. Therefore, most prior works have considered worst-case analyses and provided loose bounds on accuracy of computation as a function of precision (Castaldo et al., 2008). In contrast, the results presented in our paper were found to be tight, however they necessitate a handful of assumptions for mathematical tractability. These assumptions are listed and discussed hereafter, and the proofs of the theoretical results presented in the main text follow in this supplementary section.

**Assumption 1:** *The product terms $\{p_i\}_{i=1}^n$ are statistically independent, zero-mean, and have the same variance $\sigma_p^2$.*

This assumption, which was mentioned in the main text, is a standard one in works where the issue of variance in deep learning is studied (He et al., 2015). Note that as a result we have $Var(s_n)_{\text{ideal}} = \mathbf{E}_{\text{ideal}}\left[s_n^2\right]$ because $\mathbf{E}_{\text{ideal}}\left[s_n\right] = 0$.

**Assumption 2:** *Computation in reduced precision floating-point arithmetic is unbiased with respect to the baseline.*

This assumption is also standard in works studying quantization noise and effects (Sakr et al., 2017). An important implication is that $Var(s_n)_{\text{swamping}} = \mathbf{E}_{\text{swamping}}\left[s_n^2\right]$ because $\mathbf{E}_{\text{swamping}}\left[s_n\right] = \mathbf{E}_{\text{ideal}}\left[s_n\right] = 0$.

**Assumption 3:** *The accumulation is monotonic in the iterations leading to a full swamping event.*

This assumption means that we shall focus on a typical scenario where the partial sums $\{s_i\}_{i=1}^n$ grow in magnitude while product terms $\{p_i\}_{i=1}^n$ are of the same order. In other words, we do not consider catastrophic events where full swamping occurs unexpectedly (the probability of such event is small in any case).

**Assumption 4:** *We consider a partial sum $s_i$ that experiences full swamping in reduced precision accumulation to be statistically independent from prior partial sums $s_{i'}$ for $i' < i$.*

All partial sums $\{s_i\}_{i=1}^n$ are statistically dependent as they are computed in a recursive manner. Our assumption is that should swamping noise be so significant to cause full swamping, then the recursive dependence on prior partial sums is broken.

**Assumption 5:** *Once a full swamping event occurs, the computation of partial sum accumulation is halted.*

It is possible, but unlikely, that the computation might recover from swamping. A partial recovery of the computation is also possible but causes negligible effects on the final result. Thus, such scenarios are neglected. Assumptions 3, 4, and 5 will be particularly useful for mathematical tractability in the proof of Lemma 1.

**Assumption 6:** *The bits of the mantissa representation of partial sums $\{s_i\}_{i=1}^n$ and product terms $\{p_i\}_{i=1}^n$ are equally likely to be zero or one.*

This is yet again a standard assumption in quantization noise analysis which will be particularly useful in the proof of Theorem 1.

# A  PROOF OF LEMMA 1

In order to compute the VRR, we first need to compute $Var(s_n)$ during swamping, i.e., $Var(s_n)_{\text{swamping}} = \mathbf{E}_{\text{swamping}}\left[s_n^2\right]$, where the equality holds by application of Assumptions 1 and 2. To do so, we rely on the Law of Total Expectation. Indeed, assume that $\mathcal{A}$ is the set of events that describe all manners in which the accumulation $s_n$ experiencing swamping can occur, and let $P(A)$ be the probability of event $A \in \mathcal{A}$. Hence, by the Law of Total Expectation, we have that

$$Var(s_n)_{\text{swamping}} = \sum_{A \in \mathcal{A}} \mathbf{E}\left[s_n^2 \Big| A\right] P(A) \tag{7}$$

It is in fact a difficult task to enumerate and describe all events in $\mathcal{A}$. Thus we consider a reduced set of events $\hat{\mathcal{A}} \subset \mathcal{A}$ which is representative enough of the manners in which the accumulation occurs, yet tractable so that it can be used as a surrogate to $\mathcal{A}$ in (7). We shall form the set $\hat{\mathcal{A}}$ by application of Assumption 3, 4, and 5 as we proceed.

We consider a scenario where the first occurrence of full swamping is at iteration $i$ of the summation for $i = 2 \ldots n - 1$. This happens if:

$$|s_i| > 2^{m_{acc}} |p_{i+1}| \quad \& \quad |s_{i'}| < 2^{m_{acc}} |p_{i'+1}| \text{ for } i' = 1 \ldots i - 1. \tag{8}$$

Instead of looking at the actual absolute value of an incoming product term, we replace it by its typical value of $\sigma_p$. Furthermore, we simplify the condition of no swamping prior to iteration $i$ by only considering the accumulated sum at the previous iteration. This is due to Assumption 3 which allows us to consider a simplified scenario where the accumulation is monotonic in the iterations leading to full swamping. Hence, our simplified condition for the first occurrence of full swamping at iteration $i$ is given by:

$$|s_{i-1}| < 2^{m_{acc}} \sigma_p < |s_i|.$$

As iteration $i$ corresponds to a full swamping event, we may invoke Assumption 4 and treat each of the two inequalities above independently. Finally, we invoke Assumption 5: since full swamping happens at itertion $i$, then the result of the accumulation is $s_n = s_i$ since the computation in the following iterations is halted.

Thus, the event set $\hat{\mathcal{A}}$ we construct for our analysis consists of the mutually exclusive events $\{A_i\}_{i=2}^{n-1}$ where $A_i$ is the event that full swamping occurs for the first time at iteration $i$ under the above assumptions. The condition for event $i$ to happen is given by (8). By Central Limit Theorem, which we use by virtue of the $s_i$ being a summation of independent, indentically distributed product terms, we have that $s_i \sim \mathcal{N}(0, i\sigma_p^2)$, so that:

$$P(A_i) = 2Q\left(\frac{2^{m_{acc}}}{\sqrt{i}}\right)\left(1 - 2Q\left(\frac{2^{m_{acc}}}{\sqrt{i-1}}\right)\right) = q_i. \tag{9}$$

Furthermore, by Assumption 5 we have:

$$\mathbf{E}\left[s_n^2 \Big| A_i\right] = \mathbf{E}\left[s_i^2\right] = i\sigma_p^2. \tag{10}$$

We also add to our space of events, the event $A_n$ where no full swamping occurs over the course of the accumulation. This event happens if $|s_n| < 2^{m_{acc}} \sigma_p$ and thus has probability $P(A_n) = 1 - 2Q\left(\frac{2^{m_{acc}}}{\sqrt{n}}\right) = \tilde{q}_n$. Since this event corresponds to the ideal scenario, we have $\mathbf{E}\left[s_n^2 \Big| A_n\right] = n\sigma_p^2$.

Thus, under the above conditions we have:

$$Var(s_n)_{\text{swamping}} = \frac{1}{k}\sum_{i=2}^{n} \mathbf{E}\left[s_n^2 \Big| A_i\right] P(A_i) = \frac{\sigma_p^2}{k}\left(\sum_{i=2}^{n-1} iq_i + n\tilde{q}_n\right) \tag{11}$$

where $k = \sum_{i=2}^{n-1} q_i + \tilde{q}_n$ is a normalization constant needed as $\hat{\mathcal{A}}$ does not describe the full probability space.

Consequently we obtain (1) as formula for the VRR completing the proof for Lemma 1.

## B   PROOF OF THEOREM 1

First, we do not change the description of the events $\{A_i\}_{i=2}^{n-1}$ above. Notably, by Assumption 5, once full swamping occurs, i.e., $|s_i| > 2^{m_{acc}}\sigma_p$, the computation is stuck and stops. The probability of this event is $P(A_i)$ as above. However, to account for partial swamping, we alter the result of $\mathbf{E}\left[s_n^2\middle|A_i\right]$. Indeed, partial swamping causes additional loss of variance. When the input product terms have $m_p$ bits of mantissa, then before event $A_i$ can occur, the computation should go through each of the $m_p$ stages described in Figure 4. We again use Assumption 3 and consider a typical scenario for each of the $m_p$ stages of partial swamping preceding a full swamping event whereby the accumulation is monotonic and the magnitude of the incoming product term is close to its typical value $\sigma_p$. Under this assumption, stage $j$ is expected to happen for the following number of iterations:

$$N_j = 2^{m_{acc}+1-(m_p-j)} = 2^{m_{acc}-m_p+j+1} \tag{12}$$

for $j = 1 \ldots m_p$. At stage $j$, $j$ least significant bits in the representation of the incoming product term are truncated (swamped). The variance lost because of this truncation, which we call fractional variance loss $\mathbf{E}[f_j^2]$, can be computed by assuming the truncated bits are equally likely to be 0 or 1 (Assumption 6), so that:

$$\mathbf{E}[f_j^2] = \sigma_p^2 \left[\sum_{k=0}^{2^j-1} \frac{1}{2^j}\left(2^{-m_p}k\right)^2\right]$$

$$= \sigma_p^2 2^{-2m_p} \frac{(2^j-1)(2^{j+1}-1)}{6} \tag{13}$$

Hence, the total fractional variance lost before the occurrence of even $A_i$ is $N_j\mathbf{E}[f_j^2]$. Thus, we update the value of variance conditioned on $A_i$ as follows:

$$\mathbf{E}\left[s_n^2\middle|A_i\right] = \left(i\sigma_p^2 - N_j\mathbf{E}[f_j^2]\right)_+$$

$$= \sigma_p^2\left(i - \frac{2^{m_{acc}-3m_p}}{3}\sum_{j=1}^{m_p}2^j(2^j-1)(2^{j+1}-1)\right)_+ \tag{14}$$

where we used the operator $(x)_+ = \begin{cases} x \text{ if } x > 0 \\ 0 \text{ otherwise} \end{cases}$ in order to guarantee that the variance is positive. Effectively, we neglect the events $A_i$ where $i$ is so small that the variance retained is less than the variance lost due to partial swamping. In other words, an event whereby full swamping occurs very early in the accumulation is considered to have zero probability and we replace $P(A_i)$ in (9) by:

$$P(A_i) = q_i\mathbf{1}_{\{i>\alpha\}} \tag{15}$$

where $q_i = 2Q\left(\frac{2^{m_{acc}}}{\sqrt{i}}\right)\left(1 - 2Q\left(\frac{2^{m_{acc}}}{\sqrt{i-1}}\right)\right)$ as in (9), $\mathbf{1}$ is the indicator function, and $\alpha = \frac{2^{m_{acc}-3m_p}}{3}\sum_{j=1}^{m_p}2^j(2^j-1)(2^{j+1}-1)$ as in (14).

In addition, some boundary conditions need to be accounted for. These include the cases when no full swamping happens before the accumulation is complete but partial swamping does happen. We again consider a typical scenario as above and append our event set $\mathcal{A}$ with $m_p - 1$ boundary events $\{A'_{j_r}\}_{j_r=2}^{m_p}$, where the event $A'_{j_r}$ corresponds to the case where the computation has gone through stage $j_r - 1$ of partial swamping but has not reached stage $j_r$ yet. The condition for this event is $\sigma_p 2^{m_{acc}-m_p+j_r-1} < |s_n| < \sigma_p 2^{m_{acc}-m_p+j_r}$ and occurs typically for up to $N_{j_r-1}$ iterations. The total fractional variance lost is:

$$\sigma_p^2\left[\sum_{j=1}^{j_r-1}N_j\sum_{k=0}^{2^j-1}\frac{1}{2^j}\left(2^{-m_p}k\right)^2\right] = \sigma_p^2\frac{2^{m_{acc}-3m_p}}{3}\sum_{j=1}^{j_r-1}2^j(2^j-1)(2^{j+1}-1) \tag{16}$$

Hence,

$$\mathbf{E}\left[s_n^2 \middle| A'_{j_r}\right] = \sigma_p^2 \left(n - \alpha_{j_r}\right)_+ \tag{17}$$

where $\alpha_{j_r} = \frac{2^{m_{acc}-3m_p}}{3} \sum_{j=1}^{j_r-1} 2^j (2^j - 1)(2^{j+1} - 1)$. And the probability of event $A'_{j_r}$ is given by:

$$P(A'_{j_r}) = q'_{j_r} \mathbf{1}_{\{n > \alpha_{j_r}\}} \tag{18}$$

where $q'_{j_r} = N_{j_r-1} 2Q\left(\frac{2^{m_{acc}-m_p+j_r-1}}{\sqrt{n}}\right)\left(1 - 2Q\left(\frac{2^{m_{acc}-m_p+j_r}}{\sqrt{n}}\right)\right)$, the multiplication by $N_{j_r-1}$ reflects the number of iterations the event may occur for.

Finally, the event $A_n$ is updated and corresponds to the case where neither partial nor full swamping occurs. The condition for this event is $|s_n| < 2^{m_{acc}-m_p+1}$ and has a probability $P(A_n) = 1 - 2Q\left(\frac{2^{m_{acc}-m_p+1}}{\sqrt{n}}\right)$.

Putting things together, we use the law of total expectation as in (7) to compute:

$$Var(s_n)_{\text{swamping}} =$$
$$\frac{\sigma_p^2}{k}\left[\sum_{i=2}^{n-1}(i-\alpha)_+ q_i \mathbf{1}_{\{i>\alpha\}} + \sum_{j_r=2}^{m_p}(n - \alpha_{j_r})_+ q'_i \mathbf{1}_{\{n>\alpha_{j_r}\}} + nk_3\right] \tag{19}$$

where $k = k_1 + k_2 + k_3$, $k_1 = \sum_{i=2}^{n-1} P(A_i)$, $k_2 = \sum_{j_r=2}^{m_p} P(A'_{j_r})$, and $k_3 = P(A_n)$. Hence, the formula for the VRR in (2) in the theorem follows and this concludes the proof.

## C    PROOF OF COROLLARY 1

Applying the above analysis, we may compute the variance of the intermediate results as $\sigma_p^2 n_1 VRR(m_{acc}, m_p, n_1)$. To compute the variance of the final result, first note that the mantissa precision of the incoming terms to the inter-chunk accumulation (the results from the intra-chunk accumulation) is $\min(m_{acc}, m_p + \log_2(n_1))$. The reason being that since the intra-chunk accumulation uses $m_{acc}$ mantissa bits, the mantissa cannot grow beyond $m_{acc}$ due to the rounding nature of the floating-point accumulation. However, if $m_{acc}$ is large enough and $n_1$ is small enough, it is most likely that the mantissa has not grown to the maximum. Assuming accumulation of terms having statistically similar absolute value as was done for the VRR analysis, then the bit growth of the mantissa is logarithmic in $n_1$ and starts at $m_p$.

Hence, the variance of the computed result $s_n$ when chunking is used is:

$$Var(s_n)_{\text{chunking}} = \sigma_p^2 n_1 VRR(m_{acc}, m_p, n_1)$$
$$\times n_2 VRR\left(m_{acc}, \min\left(m_{acc}, m_p + \log_2(n_1)\right), n_2\right) \tag{20}$$

and hence the VRR with chunking can be computed using (3) in Corollary 1. This completes the proof.

