# OpenReview forum: "Accumulation Bit-Width Scaling For Ultra-Low Precision Training Of Deep Networks"
_ICLR.cc/2019/Conference_

### Official Review · AnonReviewer2 · 2018-10-27
**Clever analysis of quantization in matrix multiplies leads to actionable insights**

**Rating:** 7
**Confidence:** 4

**Review:**

There has been a lot of work on limited precision training and inference for deep learning hardware, but in most of this work, the accumulators for the multiply-and-add (FMA) operations that occur for inner products are chosen conservatively or treated as having unlimited precision. The authors address this with  an analytical method to predict the number of mantissa bits needed for partial summations during the forward, delta and gradient computation ops for convolutional and fully connected layers. They propose an information theoretic approach to argue that by using fewer bits of mantissa in the accumulator than necessary, the variance of the resulting sum is less than what it would have been if sufficient bits of mantissa were used. This is surprising to me, as quantization is usually modeled as _adding_ noise, leading to an _increase_ in variance (Mc Kinstry et al. 2018), so this is a nice counterexample to that intuition. Unfortunately the result is presented in a way that implies the variance reduction is what causes the degradation in performance, while obviously (?) it's just a symptom of a deeper problem. E.g., adding noise or multiplying by a constant to get the variance to where it should be, will not help the network converge. The variance is just a proxy for lost information. The authors should make this more clear.

Loss of variance is regarded as a proxy to the error induced/loss of information due to reduced mantissa prevision. The authors present their metric called Variance Retention Ratio (VRR) as a function of the mantissa length of product terms, partial sum (accumulator) terms, and the length of the accumulation. Thereafter, the mantissa precision of the accumulator is predicted to maintain the error of accumulation within bounds by keeping the VRR as close to 1 as possible. The authors use their derived formula for VRR to predict the minimum mantissa precision needed for accumulators for three well known networks: AlexNet, ResNet 32 and ResNet 18. For tightness analysis they present convergence results while perturbing the mantissa bits to less than those predicted by their formula, and show that it leads to more than 0.5% loss in the final test error of the network.

Some questions that the manuscript leaves open in it's current form:

0. Does this analysis only apply to ReLu networks where all the accumulated terms are positive? Would a tanh nonlinearity, e.g. in an RNN, result in a different kind of swamping behavior? I don't expect the authors to add a full analysis for the RNN case if it's indeed different, but it would be nice to comment on it.
1. Do the authors assume that the gradients and deltas will always be within the exponent range of representation? I do not find a mention of this in the paper. In other words, are techniques like loss scaling, etc. needed in addition? Other studies in literature analyzing IEEE fp16 seem to suggest so.
2. The authors do not provide details on how they actually performed the experiments when running convergence experiments. It is not straightforward to change the bit width of the accumulator mantissa in CPU or GPU kernel libraries such as CUDNN or Intel MKL. So how do they model this?
3. On page 7, the authors point out that they provide a theoretical justification of why the chunk size should neither be too small or too large - but I do not see such a justification in the paper. More detailed explanation is needed.

There are a few minor typos at a few places, e.g.

1. Page 4: “… , there is a an accumulation length….”
2. Page 6: “…floaintg-point format…"

Some figures, notably 2 and 5, use text that is unreadably small in the captions. I know this is becoming somewhat common practice in conference submissions with strict pages limits, but I implore the authors to consider shaving off space somewhere else. Some of us still read on paper, or don't have the best eyes!

---

> ### Author Response · Authors · 2018-11-14
> **Reply to AnonReviewer2 (Part 1/2)**
>
> Dear AnonReviewer2,
>
> Thank you very much for thorough review and detailed comments! We are revising our draft and will address your concerns and suggestions. In this reply, we wish to provide answers to some of your comments:
>
>
> - On the symptoms of quantization and ‘increase vs decrease’ in variance:
>
> -- Our work actually does not contradict prior findings. Indeed, prior arts have considered representation quantization, that is to say, reducing the precision of weights/activations/gradients. It is reasonable and common to model such phenomenon by an additive noise, which statistically increases the variance. What we are looking at in our work is intermediate roundings in partial sums during the accumulation. Indeed, in our work, we fixed the precision of the representations and only reduced the precision of the accumulators for partial sum accumulation. Unlike representation quantization error, the accumulation error is dominated by swamping error.
> Basic statistics tell us that in a dot product (or a sum in general), the addition of terms causes the variance of the result to increase (with the implied assumption that the terms being added are independent (He et al., 2016)). Due to the rounding of partial sums, parts of the addition are swamped away, preventing the variance of the result to grow as expected (as illustrated in Fig. 3).
> Thus, as you have correctly pointed out, we have used the loss of variance metric as a proxy to loss of information due to reduced mantissa precision. Our reasoning is that, should this loss of info be prevented in each dot product (by assigning enough precision), then we may expect the training behavior to be similar to that of the baseline.
> We will sharpen the above messages in our revised version.

---

> > ### Author Response · Authors · 2018-11-14
> > **Reply to AnonReviewer 2 (Part 2/2)**
> >
> >
> > - Reply to question 0:
> >
> > -- In our derivation, we do not require ReLU outputs (i.e., no such assumption is made). In addition, one of the three GEMM accumulations, specifically BWD, does not involve activations (ReLU outputs); it involves activation gradients and weights. Even the other two accumulations (FWD and GRAD) accumulate elementwise products of activations and weights/activation gradients. Thus, we never deal with the case of an accumulation where all terms are positive. We do not expect a tanh non-linearity to exhibit any different behavior.
> > In case of RNN, we validated our theory with a simple example, PTB for language model. In case of PTB, the swamping issue was not severe, mainly due to its short accumulation length (i.e., for PTB-medium (minibatch=20, timestep=35, the accumulation length for FWD,BWD,WGRAD are only 650, 2600, and 700, respectively). Thus, for this network topology, VRR predicted 5-bit for accumulation (with chunking) which is much smaller than one used for CIFAR10. We demonstrated successful convergence using this precision of accumulation, as shown in this anonymous link: https://www.dropbox.com/s/yntvzhnvso64z29/ptb.pdf?dl=0
> >
> > It is to be noted that, a more difficult task, say WMT, typically requires a much larger batch size of a few thousand, so that the GRAD accumulation length could reach a million. This would make such example very interesting and relevant to our work. Unfortunately, we currently do not have a working setup for training an LSTM on such a task. As we mentioned in the conclusion, this is definitely a topic of our future work. Nevertheless, we will include comments on this issue in our revised version as per your request.
> >
> >
> > - Reply to question 1:
> >
> > -- Yes, we do make this assumption and solely analyze the mantissa precision requirements. This is mentioned at the start of Section 4. In addition, we do use the technique of loss scaling in our experiments. We just realized that in our submitted draft, we somehow omitted to mention that. We are using the same technique of loss scaling as Micikevicius et al. (2017) and Wang et al. (2018) for the (1,5,2) representation and should have mentioned that at the start of Section 5. Thanks a lot for this question! We will add this mention in the revised version.
> >
> >
> > - Reply to question 2:
> >
> > -- Very good point. We have used an in-house library to perform the experiments and we could not shed too much light on those details. However, we can answer with the crucial part needed to reproduce the results, which in principle is applicable to any deep learning framework. The key is to modify the CUDA code of the GEMM function. In particular, there is a for loop where the partial sum accumulation occurs. There, we add a call to a custom rounding function (which quantizes the partial sum to the desired reduced precision floating-point representation). We will add an explanation such as this in the revised version.
> >
> >
> > - Reply to question 3:
> >
> > -- The justification is actually mentioned in the last paragraph of Section 4 when discussing Fig. 5 (c). Indeed, the “plateaus” in the VRR curves as a function of chunk size indicate that a specific value of chunk size is not of great importance as long as it is neither too small nor too large since in the extreme cases the VRR does drop. One intuition we can provide further is that, in chunk-based accumulation, there are two sources of swamping errors: swamping in the inter-chunk accumulation and swamping the intra-chunk accumulations. If the chunk size is too small, then the inter-chunk accumulation is very similar to the original accumulation; while a very large chunk size makes the intra-chunk accumulations similar to the original one. In either case, one of the two types of accumulations would suffer the same fate as the original accumulation.
> >
> >
> > Finally, thanks for your minor comments, we will correct the typos and try to magnify the plots as much as possible. We sympathize with your comments, we also like to read on paper.

---

> ### Author Response · Authors · 2018-11-21
> **Revision Uploaded**
>
> Dear AnonReviewer2,
>
> This is a note to let you know that we have uploaded our revision. To make it easy for you to track changes, we have typed all modifications with respect to the original manuscript in blue color.
>
> In particular, at the end of Section 3 (Page 4), we have added a small paragraph to highlight even more how our study is different from mainstream research on neural network quantization. We included a short discussion on RNNs in the conclusion, as per your request. We added mentions to the technique of loss scaling used, as well as some details of the partial sum rounding implementation, both at the start of Section 5 (Page 7). Finally, we added a sentence to supplement our explanation of the choice of chunk size in the final paragraph of Section 4 (Page 7) as per our original reply to your review.
>
> We thank you again for your nice review!

---

### Official Review · AnonReviewer1 · 2018-11-02
**Interesting theoretical framework for predicting the necessary precision in deep networks, along with experimental evaluation confirming the theoretical results.**

**Rating:** 6
**Confidence:** 3

**Review:**

Quality and clarity:
The paper presents a theoretical framework and method to determine the necessary number of bits in a deep learning networks. The framework predicts the smallest number of bits necessary in the (multiply-add) calculations (forward propagation, backward propagation, and gradient calculation) in order to keep the precision at an acceptable level.

The statistical properties of the floating-point calculations form the basis for the approach, and expressions are derived to calculated the smallest number of bits based on, e.g., the length of the dot product and the number variance.

The paper seems theoretically correct, although I haven't studied the appendices in detail. The experimental part is good, using three networks of various sizes (CIFAR-10 ResNet 32, ImageNet ResNet 18 and ImageNet AlexNet) as benchmarks. The experimental results support the theoretical predictions.

Originality and significance:
The paper seems original, at least the authors claim that no such work has been done before, despite the large amount work done on weight quantization, bit reduction techniques, etc. The paper may have some significance, since most earlier papers have not considered the statistical properties of the reduced precision calculations.

Pros:
* Interesting topic
* Theoretical predictions match the practical experiments

Cons:
* Nothing particular

Minor:
* Fig 5a. The curve for m_acc = 13 does not seem to follow the same pattern as the other curves. Why?
* Motivate why you have selected the networks that you have in the evaluation.

---

> ### Author Response · Authors · 2018-11-14
> **Reply to AnonReviewer1**
>
> Dear AnonReviewer1,
>
> Thank you very much for your nice review! We are revising our draft to take into account all comments and suggestions. In this reply, we wish to provide a response to some of your comments:
>
>
> - Reply to the comment on originality and significance:
>
> -- We just want to mention that, indeed, there is a large body of work addressing the general problem of quantization and reduced precision in deep learning. Almost all of these works solely focus on the issue of representation quantization (i.e., reducing the precision of weights and/or activations). To this day, the precision of partial sums in accumulations has been largely overlooked. Hence, this is still an unanswered question, and as described in our introduction (third paragraph and Fig. 1 (b)), an important one to address in order to scale down the hardware complexity of deep learning systems. This constitutes the thesis of our work and is why we have claimed that no such work has been done before, and why our paper is of great significance. Of course, that, in addition to the statistical analysis, which by itself is novel.
>
>
> - Reply to first minor comment:
>
> -- We also noticed the peculiar pattern around m_acc = 12 & 13 in Fig. 5 (a). Observe that this curve is obtained by evaluation of eq. (2) in Theorem 1, which is clearly a non-linear function of m_acc. We can actually give you a more elaborated answer. We have plotted v(n) as a function of m_acc for several fixed values of n (as in our paper, we used a value of m_p=5). Please check out the plot at this anonymous link: https://www.dropbox.com/s/au9h9v650dvhyxw/variance_lost_fixed_n.pdf?dl=0
> These plots are for illustrative purpose, note that a fractional value of m_acc has no physical meaning. As we can see, the general trend of variance lost decreasing as a function of m_acc is present (which is expected). However, there is a ‘lobe’ in each curve. This is because there are two sources of errors modeled by eq. (2): full swamping errors (contributing to the equation via the first term in the numerator), and partial swamping errors (contributing to the equation via the second term in the numerator). Thus, there is a trade-off between these two sources of errors: in some regime, full swamping would dominate (when m_acc is much higher than m_p) while in another regime partial swamping would dominate (when m_acc is not much higher than m_p). This trade-off causes the ‘bumps’ or ‘lobes’ observed in our linked plot. The effects seem most dramatic for values of m_acc around 12 and 13 which explains the pattern observed in Fig. 5 (a).
>
>
> -Reply to second minor comment:
>
> -- There are two reasons why we have selected our benchmarks. First, the datasets and networks are widely used and popular in such applications. Second, such image datasets and associated convolutional networks presents very large accumulation lengths due to the data size and network topology. This makes them very good candidates against which we can verify our work.

---

> ### Author Response · Authors · 2018-11-21
> **Revision Uploaded**
>
> Dear AnonReviewer1,
>
> This is a note to let you know that we have uploaded our revision. To make it easy for you to track changes, we have typed all modifications with respect to the original manuscript in blue color.
>
> In particular, at the end of Section 3 (Page 4), we have added a small paragraph to highlight even more how our study is different from mainstream research on neural network quantization. Furthermore, at the start of Section 5 (Page 7), we include a sentence motivating the choice of the benchmarks.
>
> We thank you again for your nice review!

---

### Official Review · AnonReviewer3 · 2018-11-05
**Accumulation Bit-Width Scaling For Ultra-Low Precision Training Of Deep Networks**

**Rating:** 6
**Confidence:** 4

**Review:**

The authors conduct a thorough analysis of the numeric precision required for the accumulation operations in neural network training. The analysis is based on Variance Retention Ratio (VRR), and authors show the theoretical impact of reducing the number of bits in the floating point accumulator. And through extensive benchmarks with popular vision models, the authors demonstrate the practical performance of their theoretical analysis.

There are several points that I am not particularly clear about this work:

1) In section 4.4, the authors claim to use v(n) < 50 as the cutoff of suitability. This is somewhat arbitrary. As one can imagine, for an arbitrary model of VRR, we can find an empirical cutoff that seems to match benchmarks tightly. Or put it another way, this is a hyperparameter that the authors can tune to match their chosen benchmarks. It would be more interesting to see a detailed study on this cutoff on multiple datasets.

2) Again the 0.5% accuracy cutoff from baseline in the experiment section is also another similar hyperparameter.

It would be more convincing if we can see a fuller picture of the training dynamics without these two hyperparameters clouding the big picture.

Having said this, I appreciate the authors' effort in formally studying this problem.

---

> ### Author Response · Authors · 2018-11-14
> **Reply to AnonReviewer3**
>
> Dear AnonReviewer3,
>
> Thank you very much for your review and thoughtful comments. Those will be addressed as we prepare our revised draft. In the meantime, we would like to provide a first response:
>
>
> - Response to point 1)
>
> -- In Fig 5 (a&b), the variance lost rapidly increases when v(n)>50 and n increases. On the other hand, when v(n)<50 and n decreases, the variance lost quickly drops to zero. As such, v(n)=50 coincides with the ‘knee’ of the variance lost curve as a function of accumulation length and was therefore chosen as stability cutoff. Thus, this choice of cutoff was chosen purely based on the accumulation length and precision, independently from the benchmarks we have used.
>
>
> - Response to point 2)
>
> -- It is believed that there is an error bound for neural networks by changing the random number seeds. In ref [a], table 1 shows that the random seed effect could achieve 0.56% in ImageNet and in ref [b], page 5, the difference of random seed effect can be 0.44% in CIFAR100. In our experiments, we also observed the a variability of ~0.5% in accuracy due to random seed variation on CIFAR-10 and ImageNet.  In this paper, we used this well-accepted value for comparison to a baseline. In addition, choosing this value is mainly for illustration purpose and won’t change the conclusion of our work. For example, in Fig 6 (d), using the predicted precision assignment, the converged test error can be clearly seen to be very close to the baseline, but increases significantly when the precision is further reduced.
>
>
> We will motivate the choice of these two numbers more explicitly in the revised draft. We do want to let you know that we appreciate your comment, and we also believe a story is better told when specific numbers do not ‘cloud’ the picture. Hopefully our justification above makes it more convincing.
>
>
> Thanks again for your review!
>
>
> [a] Goyal et al. (2018), Accurate, Large Minibatch SGD: Training ImageNet in 1 Hour - https://arxiv.org/pdf/1706.02677.pdf
> [b] Gastaldi (2017), Shake-Shake regularization - https://arxiv.org/pdf/1705.07485.pdf

---

> ### Author Response · Authors · 2018-11-21
> **Revision Uploaded**
>
> Dear AnonReviewer3,
>
> This is a note to let you know that we have uploaded our revision. To make it easy for you to track changes, we have typed all modifications with respect to the original manuscript in the blue color.
> In particular, we have further motivated the choice of the two numbers of v(n) and accuracy cut-off on pages 6 and 8, respectively.
>
> We thank you again for your nice review!

---

### Meta-Review · Area_Chair1 · 2018-12-13

**Confidence:** 4
**Recommendation:** Accept (Poster)

**Metareview:**

The authors present a theoretical and practical study on low-precision training of neural networks. They introduce the notion of variance retention ratio (VRR) that determines the accumulation bit-width for
precise tailoring of computation hardware.  Empirically, the authors show that their theoretical result extends to practical implementation in three standard benchmarks.

A criticism of the paper has been certain hyperparameters that a reviewer found to be chosen rather arbitrarily, but I think the reviewers do a reasonable job in rebutting it.

Overall, there is consensus that the paper presents an interesting framework and does both practical and empirical analysis, and it should be accepted.